# Gene Therapy for BCG-Unresponsive Non-Muscle Invasive Bladder Cancer: Current Evidence and Future Directions

**DOI:** 10.3390/cancers17223631

**Published:** 2025-11-12

**Authors:** Philippe Pinton

**Affiliations:** Ferring Pharmaceuticals, Ferring International PharmaScience Centre, Amager Strandvej, 2770 Kastrup, Denmark; phpi@ferring.com

**Keywords:** gene therapy, urothelial carcinoma, non-muscle invasive bladder cancer, carcinoma in situ

## Abstract

Some bladder cancers do not respond to the usual *Bacillus Calmette-Guérin* (BCG) treatment, leaving patients with few options, often requiring complete bladder removal. This review looks at gene therapy as a different way to treat these cases while keeping the bladder intact. Gene therapy works by delivering helpful genetic material directly into the bladder to fight cancer cells and boost the body’s immune response. Three gene therapies (nadofaragene firadenovec, cretostimogene grenadenorepvec, and detalimogene voraplasmid) are discussed. These treatments might provide bladder-preserving alternatives for patients who have limited options.

## 1. Introduction

Bladder cancer is the ninth most prevalent cancer globally [1], with a higher incidence in men and a median diagnosis age of 73 years [2]. Smoking remains the predominant risk factor, alongside old age and chemical exposure [3,4]. Haematuria is the most frequent presenting symptom leading to detection [4]. Over 90% of bladder cancers are urothelial carcinomas, encompassing several histological variants, including micropapillary, nested/microcystic, plasmacytoid, and squamous-glandular subtypes [4]; nonurothelial carcinomas, such as squamous cell carcinoma, adenocarcinoma or carcinoma of other cell origins, are comparatively rare [3,4]. Urothelial carcinoma can be classified based on the depth of muscle invasion as non-muscle invasive bladder cancer (NMIBC) or muscle invasive bladder cancer (MIBC), where the former is confined to the inner lining of the bladder and makes up approximately 70% of new diagnoses [5,6].

Risks of recurrence or progression in NMIBC may be substantial depending on the tumour stage (Figure 1) [7]. Ta papillary tumours, confined to the innermost layer of the bladder lining, are considered low-grade if less aggressive or high-grade if more aggressive (i.e., growing faster, spreading more readily) [8]. Low-grade Ta tumours exhibit high recurrence rates (15–70% at one year) but a relatively low likelihood of progression to MIBC (<5%), whereas high-grade Ta tumours demonstrate increased risk of progression to inner lining invasion (13–40%) or MIBC (6–25%) [7,9]. In stage T1, where tumours invade the inner lining and connective tissue, risk of recurrence is approximately 80%, with a considerable probability of progression (50%) within three years [7,8,9]. Carcinoma in situ (CIS) refers to flat, non-invasive, high-grade tumours that are associated with high rates of both recurrence (82%) and progression (42–83%), especially if not treated with intravesical therapy [7,8].

The current standard of care for NMIBC involves transurethral resection of bladder tumour (TURBT), followed by adjuvant intravesical therapy, usually with immunostimulant *Bacillus Calmette-Guérin* (BCG) [5]. Despite interventions with TURBT and BCG, significant unmet needs persist due to disease recurrence in approximately 50% of patients, or non-responsiveness to BCG in approximately 33% of all patients with NMIBC [10,11]. BCG-unresponsive was defined by the U.S. Food and Drug Administration (FDA) as persistent or recurrent CIS alone or with recurrent Ta/T1 disease within 12 months of completion of adequate BCG therapy; recurrent high-grade Ta/T1 disease within six months of completion of adequate BCG therapy; or T1 high-grade disease at the first evaluation following an induction BCG course [12]. The persistent global BCG shortage has led to guideline updates that reserve BCG treatment for intermediate or high-risk patients [2,13,14,15].

Radical cystectomy, the complete removal of the bladder, remains the recommended option after BCG unresponsiveness (Figure 2) [6]; however, it may not be viable due to concerns revolving its invasiveness, significant morbidity and permanent impact on quality of life (QoL). In addition, patients may be unwilling to undergo extirpative surgery [6,16]. Alternative treatment options, as recommended by the National Comprehensive Cancer Network guidelines, include intravesical chemotherapy with the approved valrubicin, or pembrolizumab for select patients (i.e., BCG-unresponsive, high-risk NMIBC with CIS who are ineligible for or have elected not to undergo cystectomy) [5]. However, effective treatment options remain sparse as a 10% 12-month recurrence-free survival rate was reported for valrubicin [17].

In Japan, the treatment landscape for high-risk NMIBC is shaped by unique clinical practices and cultural considerations. While the Japanese Urological Association (JUA) guidelines recommend both induction and maintenance BCG therapy [18], real-world data have shown that most patients receive only six to eight doses of induction therapy, with maintenance therapy being infrequently used [18,19]. This is partly due to the perception among Japanese urologists that extended induction (e.g., eight doses) may be equivalent to maintenance, and partly due to concerns about adverse events (AEs), cost, and BCG shortages [19]. As a result, only 17% of patients receive maintenance BCG, and just 8% undergo a second induction [20]. These patterns highlight a significant gap between guideline recommendations and clinical practice, underscoring the need for alternative, bladder-sparing therapies.

Gene therapy involves introducing nucleic acid into host cells to elicit therapeutic effects via several mechanisms, such as repairing mutated tumour suppressor functions or modulating the host’s immune response to produce anti-tumour activities [5]. While gene therapy has a wide range of applications in medicine, current research efforts focus on utilising it in the treatment of human cancers [5,11]. Various vectors have been used in clinical trials, including adenoviral, retroviral, herpes virus, and naked plasmid vectors [11]. Among these options, adenoviruses are often preferred due to their ability to express therapeutic genes episomally, meaning the genetic material remains separate from the host genome, thereby reducing the risk of insertional mutagenesis and associated oncogenic events [11]. The bladder’s anatomy, consisting of a cavity that allows for direct contact between the vector with a therapeutic gene and the tumour, makes it an ideal target for gene therapy development [5]. The relatively easy access to urine and tissue samples facilitates the monitoring of therapeutic effect [5]. Multiple gene therapy candidates have been investigated for the treatment of bladder cancer in the past decade. In December 2022, the FDA approved the first gene therapy, nadofaragene firadenovec, for high-risk BCG-unresponsive NMIBC with CIS [11,21]. Other gene therapies, such as crestostimogene grenadenorepvec (CG0070) and detalimogene voraplasmid (EG-70) are currently under investigation.

This article presents an overview of considerations for gene therapy development, current gene therapies indicated for high-risk BCG-unresponsive NMIBC with CIS (Table 1) and explores future research directions aimed at expanding the therapeutic armamentarium against this malignancy.

## 2. Considerations for Gene Therapy in Bladder Cancer

The development and implementation of gene therapies for bladder cancer face several challenges. First, stable and efficient gene vectorisation is critical, as viral vectors must reliably deliver therapeutic genes into urothelial cells while maintaining transduction efficiency [28]. Ensuring sustained gene expression is equally essential to produce the desired protein in sufficient quantities to elicit therapeutic effects. Viral safety is a concern as vectors may present potential risks, such as unwanted immune responses or AEs. The local intravesical administration route requires effective and reproducible delivery methods to maximise vector–tumour contact while limiting systemic exposure. In addition, the large-scale production of gene therapeutics requires stringent control over quality and purity, which can be complex and costly [28]. Regulatory barriers, such as those related to chemistry, manufacturing, and controls may be cumbersome, leading to delays in development and impact on payer perception. Addressing these multifaceted challenges is crucial to advance gene therapy as a strategic tool in the treatment of NMIBC.

## 3. Nadofaragene Firadenovec

Nadofaragene firadenovec is an FDA-approved, nonreplicating adenoviral vector-based gene therapy developed for BCG-unresponsive NMIBC (Table 1) [21,29]. This therapy is instilled into the bladder once every three months, with a one-hour dwell time [11,21,29].

A gene transfer enhancing agent, Syn3, is used with nadofaragene firadenovec to facilitate its entry into uroepithelial cells. Once inside the cells, the adenoviral vector delivers the interferon alpha-2b transgene (*IFNα2b*) to the nucleus. Subsequent transcription and translation result in sustained local production of IFNα2b protein, which exerts both direct cytotoxic and indirect pleiotropic anti-tumour effects (Figure 3a) [11]. Direct mechanisms include the induction of endoplasmic reticulum stress leading to caspase-mediated apoptosis, upregulation of tumour necrosis factor-related apoptosis-inducing ligand (*TRAIL*) expression promoting cell death, inhibition of angiogenesis via downregulation of growth factors such as bFGF causing tumour hypoxia and central necrosis, and enhanced tumour antigen presentation through increased major histocompatibility complex class 1 (MHC-I) expression [11]. Indirectly, IFNα stimulates dendritic cell priming of cytotoxic CD8+ T cells and enhances natural killer cell activity against tumour cells lacking MHC-I [11].

Preclinical studies of recombinant adenovirus expressing *IFNα* along with gene transfer enhancing agent *Syn3* (rAd-IFNα/Syn3) demonstrated tumour regression of bladder cancer in murine models while achieving sustained high levels of IFNα in both urine and urothelium [30,31,32]. These results supported progression to a phase I study (NCT01162785), which confirmed safety and tolerability, with micturition urgency as the most common AE (88% of 17 patients); no dose-limiting toxicity was observed [33]. A subsequent phase Ib study (NCT01162785) evaluated whether a second instillation would enhance therapeutic outcomes; however, findings indicated that a single instillation was sufficient to achieve adequate urinary IFNα concentrations [34]. A phase II study (NCT01687244) of rAd-IFNα/Syn3 continued to show a favourable safety profile and promising efficacy. Of the 40 patients receiving rAd-IFNα/Syn3, 35% (n = 14) achieved 12-month high-grade recurrence-free survival. No Grade 4 or 5 AEs occurred; the most frequently reported AEs related to the drug included micturition urgency (40%), dysuria (40%), and fatigue (32.5%) [13]. In the pivotal phase III study (NCT02773849), nadofaragene firadenovec demonstrated a complete response (CR) rate of 53.4% within three months post-treatment, which was maintained in 45.5% of patients at 12 months (Table 2) [35]. Long-term follow-up over five years indicated durable treatment response, including bladder preservation in nearly half of the treated patients and a CR rate of 11% at 57 months, without new safety concerns emerging [22,36].

Safety analyses across all study phases showed that AEs were predominantly mild and localised, typically lasting less than two days. Grade 3 drug-related AEs occurred in 3.8% of patients, with no Grades 4 or 5 drug-related events reported. Treatment discontinuation due to AEs was uncommon (1.9%) [22,35,36]. Recent real-world data from the Mayo Clinic sites in the United States demonstrated a 73% CR rate at three months and a 67% CR rate at six months in patients with CIS [39]. An ongoing phase IV non-interventional study (NCT06026332) of nadofaragene firadenovec will examine further outcomes in the real-world setting [40]. Initial findings from a phase III study in Japanese patients (NCT05704244) showed 75% CR at three months; most drug-related AEs (84.2%) were Grade 1, no Grade ≥ 3 drug-related AEs were reported [41]. These findings suggest a favourable benefit-to-risk ratio of nadofaragene firadenovec in this patient population.

The potential of combination therapy of nadofaragene firadenovec with chemotherapy or immunotherapy is currently under investigation for high-grade BCG-unresponsive NMIBC [42]. Studies are also ongoing to investigate nadofaragene firadenovec in intermediate-risk NMIBC, an indication without any FDA-approved treatment at the time of writing [43], as well as in low-grade upper tract urothelial carcinoma [44].

## 4. Cretostimogene Grenadenorepvec (CG0070)

CG0070 is an investigational, engineered, conditionally replicating oncolytic adenovirus targeting the retinoblastoma (Rb) gene pathway-defective cells commonly present in urothelial carcinoma (Table 1) [23]. This therapy is administered intravesically over six weekly doses, followed by three weekly maintenance dosing at months 3, 6, 9, 12, and 18 [24]. CG0070 has received FDA Fast Track and Breakthrough Therapy designations for high-risk, BCG-unresponsive NMIBC with CIS [45].

CG0070 works in two distinct but synergistic mechanisms. The first mechanism involves inducing tumour cell lysis through selective replication (Figure 3b) [23]. Bladder cancer cells frequently exhibit a defective Rb pathway, which leads to increased expression of *E2F*, a transcription factor that regulates cell cycle progression. The abundance of E2F promotes the replication of the virus, subsequently leading to tumour cell lysis [23,24,25]. The second mechanism involves immune-mediated tumour destruction via granulocyte-macrophage colony-stimulating factor (GM-CSF) production [23,24]. As the virus replicates and causes oncolysis, it releases GM-CSF and tumour-associated antigens (TAAs), stimulating the immune system to promote immunogenic cell death [23,24].

CG0070 is under investigation as monotherapy or combination therapy. As monotherapy, CG0070 is being evaluated for both high-risk and intermediate-risk NMIBC. For high-risk BCG-unresponsive NMIBC, a phase I study (NCT00109655) established the safety and tolerability of intravesical CG0070 administration, with Grade 1 or 2 bladder toxicities as the most common AEs; no dose-limiting toxicities were observed [46]. These phase I study findings prompted further development, leading to the phase II study, BOND-002 (NCT02365818), which reported a CR rate of 47% at six months among patients with CIS ± Ta/T1 (pure CIS was 58%) [23]. Recent phase III results from the BOND-003 study (NCT04452591) reported a CR rate of 75.5% (83 of 110 patients) at any time and 46.4% (51 of 110 patients) at 12 months, with an estimated 24-month CR rate of 42.3% [37]. Study drug-related AEs were Grade 1 or 2, with a median time to resolution of one day [37]. For intermediate-risk NMIBC, a multi-national, randomised phase III PIVOT-006 study (NCT06111235) comparing CG0070 versus surveillance after TURBT, is in progress [47,48].

As combination therapy, intravesical CG0070 combined with intravenous pembrolizumab demonstrated robust CR rates of 57% (20 of 35 patients) and 83% (29 of 35 patients) at 12 months and at any time, respectively, in the phase II CORE-001 study (NCT04387461) for high-risk BCG-unresponsive NMIBC [24]. Safety profiles were consistent with the known AEs of each agent, no synergistic toxicity was observed [24]. Of relevance, the CR rate for pembrolizumab monotherapy was 41% at three months [49], while the CR rate for CG0070 monotherapy was 46% at 12 months [37]; however, it is crucial to note that the efficacy rates from these studies were not designed for direct head-to-head comparison. A separate phase II study, CORE-008 (NCT06567743), for high-risk BCG-exposed NMIBC with intravesical CG0070 has also been planned [50]. An Expanded Access Program (NCT06443944) is ongoing to provide CG0070 to real-world patients with BCG-unresponsive NMIBC with CIS who are ineligible for clinical trials [51].

## 5. Detalimogene Voraplasmid (EG-70)

EG-70 is a novel, investigational, non-integrating, non-viral gene therapy designed to elicit local stimulation of anti-tumour immune response in the bladder, while minimising the potential for off-target, systemic immune-related toxicities [26]. This non-viral delivery system carries a plasmid that expresses interleukin-12 (*IL-12*) and retinoic acid-inducible gene I (*RIG-I*), where the latter is an innate immunity regulator that recognises infected cells (Figure 3c) [10,26,27]. Its nanoparticle formulation reduces the burden of administration and decontamination procedures for patients and clinicians [26].

A preclinical study showed that EG-70 activated innate and adaptive anti-tumour immune responses by remodelling the tumour environment [27,38]. The phase I LEGEND study (NCT04752722) demonstrated that EG-70 was well-tolerated, with a CR rate of 73% in patients with NMIBC with CIS; the recommended phase II dose was identified [10,26,27,38]. The study progressed to phase II (NCT04752722), which is ongoing at the time of writing [10,26,27]. Preliminary phase II results have recently been reported, with an overall CR rate of 71% (15/21 patients), and CR rates of 67% (14/21) at three months and 47% (8/17) at six months. Treatment-related AEs (TRAEs) were all of Grade 1 or 2 severity, with the most common TRAEs including dysuria (21.4%) and bladder spasms (19.0%) [38].

Overall, the mechanisms of actions of nadofaragene firadenovec, cretostimogene grenadenorepvec, and detalimogene voraplasmid (Figure 3) highlight the diversity of gene therapy strategies and may inform the identification of biomarkers predictive of response or resistance. For example, elevated anti-adenovirus antibody titres have been associated with durable responses to nadofaragene firadenovec [52]. Similarly, Table 2 summarises key efficacy and safety outcomes that may influence treatment sequencing in clinical practice. Agents with favourable safety profiles and durable responses may be prioritised for patients seeking bladder preservation, while those with higher rates of AEs may be reserved for select populations. These practical considerations underscore the need for individualised treatment planning based on both clinical trial data and patient preferences.

## 6. Comparative Effectiveness

While gene therapy trials for BCG-unresponsive NMIBC, such as those evaluating nadofaragene firadenovec, cretostimogene grenadenorepvec, and detalimogene voraplasmid, have demonstrated promising response rates, the majority of these studies are single-arm and non-comparative in design [35,37,38]. This methodological limitation restricts the generalisability of findings and complicates direct comparisons with established standards such as radical cystectomy or systemic immunotherapy [49]. The absence of head-to-head trials means that reported efficacy and safety outcomes should be interpreted with caution, as differences in patient selection, endpoints, and follow-up may influence the results.

To address this gap, the use of synthetic (external) control arms—constructed from historical clinical trial data or real-world evidence representing standard-of-care outcomes—has emerged as a promising methodological approach [52,53]. Synthetic arms can serve as external comparators for efficacy, safety, and cost-effectiveness analyses, enabling more robust contextualisation of single-arm trial results. For example, recent cost-effectiveness analyses in BCG-unresponsive NMIBC have used synthetic arms derived from single-arm trials and real-world cohorts to compare novel therapies such as gemcitabine/docetaxel, pembrolizumab, and hyperthermic intravesical chemotherapy against radical cystectomy, the standard of care [54]. In broader oncology, synthetic control arms have been successfully constructed for non-small cell lung cancer and other indications and are increasingly accepted by regulatory agencies and health technology assessment bodies to inform decision-making when randomised controlled trials are not feasible [55,56]. These approaches enable robust comparisons of efficacy, safety, and cost-effectiveness, and are likely to play an increasing role in the evaluation of emerging therapies for NMIBC. Future research should prioritise the development and validation of high-quality synthetic control arms to enhance the interpretability and translational value of new treatments.

## 7. Future Perspectives

Current gene therapy options show considerable promise, particularly for patients with NMIBC who are unresponsive to BCG treatment. As the therapeutic landscape for bladder cancer continues to evolve, future advancements in gene therapy may focus on several key areas.

While current clinical trials have demonstrated promising efficacy and safety profiles for gene therapies in BCG-unresponsive NMIBC, most studies are single-arm and lack direct comparators. This limits the ability to contextualise outcomes against existing standards such as radical cystectomy or systemic immunotherapy. Furthermore, patient populations in trials may not fully reflect real-world diversity, particularly in terms of age, comorbidities, and prior treatment history. Future studies should aim to incorporate comparative designs and broader inclusion criteria to enhance generalisability and inform clinical decision-making.

### 7.1. Patient Selection and Biomarker Development

Optimising patient selection through biomarker-driven approaches holds potential in enhancing therapeutic outcomes. Identifying specific patient characteristics or biomarkers that predict treatment sensitivity or resistance would enable clinicians to better identify likely responders, thereby improving the overall response rates and minimising unnecessary exposure [22,53]. For example, elevated anti-adenovirus antibody titres have been correlated with durable responses to nadofaragene firadenovec [52].

Additionally, the exploration of novel, alternative vectors could move this field forward, especially with vectors that can enhance transfection efficiency to improve treatment outcomes. Recent research on lentiviral vectors has shown them to transduce stably across the bladder urothelium [57]. To expand treatment options, future research may also focus on the development of novel combination strategies that target resistance pathways [22]. For instance, programmed death-ligand 1 and epidermal growth factor receptor are potential targets that have been identified and warrant further studies when combined with interferon gene therapy [22,53].

### 7.2. Patient Perspective and QoL Considerations

Gene therapy represents a significant advancement in offering bladder-sparing treatment options to patients with BCG-unresponsive NMIBC. The recommended option post-BCG unresponsiveness, radical cystectomy, involves complete removal of the bladder and is associated with significant morbidity and declines in QoL measures [6,16,58]. Gene therapies provide an alternative with a manageable safety profile. The localised intravesical administration limits systemic exposure, reducing the risk of severe AEs commonly seen with systemic immunotherapies [6]. These factors could contribute to improved tolerability and potentially better QoL, preserving both physical function and psychological well-being [58]. As patient-centred care becomes increasingly prioritised, gene therapy may fulfil an important unmet need by balancing clinical benefits with health-related QoL.

QoL outcomes are increasingly recognised as critical endpoints in NMIBC management. Validated instruments, such as the European Organisation for Research and Treatment of Cancer (EORTC) QLQ-NMIBC24 and Functional Assessment of Cancer Therapy-Bladder-Cystectomy (FACT-Bl-Cys), have been used to assess patient-reported outcomes in bladder cancer trials. To date, there are no published data from gene therapy trials in BCG-unresponsive NMIBC that report patient-reported outcomes (PROs) or QoL using validated instruments such as the EORTC QLQ-NMIBC24 or FACT-Bl-Cys. Recent pivotal and real-world studies of nadofaragene firadenovec, cretostimogene grenadenorepvec, and detalimogene voraplasmid have focused on efficacy and safety endpoints, without reporting standardised patient-cantered measures. This represents a significant gap in the current evidence base. Incorporating validated PRO and QoL instruments in future gene therapy studies is essential to fully characterise the impact of these therapies from the patient perspective, particularly in comparison with radical cystectomy or other bladder-sparing approaches [55].

### 7.3. Cost and Access Considerations

The introduction of gene therapies into clinical practice raises important considerations around cost-effectiveness, accessibility and healthcare system implications. While gene therapies may involve complex manufacturing processes that can impact pricing, their outpatient intravesical administration could reduce hospitalisation and associated costs compared with cystectomy [59]. Preliminary health economic analyses suggest that bladder-sparing approaches may offer favourable cost-effectiveness profiles, particularly when factoring in quality-adjusted life years (QALYs) and reduced need for extensive postoperative care [54,59]. Nevertheless, comprehensive health technology assessments and real-world cost-effectiveness studies are needed to support broader adoption and inform reimbursement decisions. Beyond clinical efficacy, regulatory approval pathways and reimbursement frameworks significantly influence access to gene therapies. The complexity of manufacturing and quality control requirements may delay market entry and affect payer perceptions. In regions with limited healthcare budgets or reimbursement infrastructure, access to these therapies may be constrained. Health technology assessments and cost-effectiveness analyses will be essential to support broader adoption and equitable access.

## 8. Conclusions

Bladder cancer is well-suited for gene therapy due to its anatomical and biological characteristics. To date, nadofaragene firadenovec is the only gene therapy approved by the FDA for patients with high-risk BCG-unresponsive NMIBC with CIS, with other promising interventions, such as cretostimogene grenadenorepvec and detalimogene voraplasmid, under development. Research continues to explore ways to improve patient selection, investigate novel vectors, and evaluate combination strategies to address treatment challenges in this disease.

## Figures and Tables

**Figure 1 cancers-17-03631-f001:**
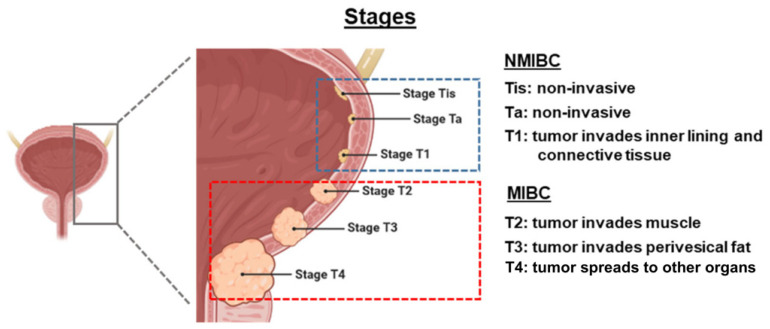
Staging of bladder cancer. Figure adapted from Herranz R et al. [8], published under the Creative Commons Attribution License (CC BY). MIBC, muscle-invasive bladder cancer; NMIBC, non-muscle invasive bladder cancer.

**Figure 2 cancers-17-03631-f002:**
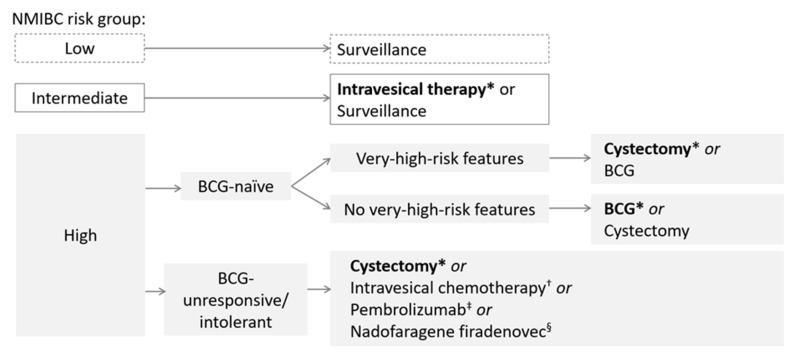
Overview of NMIBC treatment guidelines. * Indicates preferred treatment. ^†^ Valrubicin is approved for BCG-refractory CIS. ^‡^ Indicated for BCG-unresponsive, high-risk NMIBC with CIS who are ineligible for or have elected not to undergo cystectomy. ^§^ Indicated for BCG-unresponsive, high-risk NMIBC with CIS, may also be considered for BCG-unresponsive, high-risk NMIBC with high-grade papillary Ta/T1 only tumours without CIS. BCG, *Bacillus Calmette-Guérin*; CIS, carcinoma in situ; NMIBC, non-muscle invasive bladder cancer.

**Figure 3 cancers-17-03631-f003:**
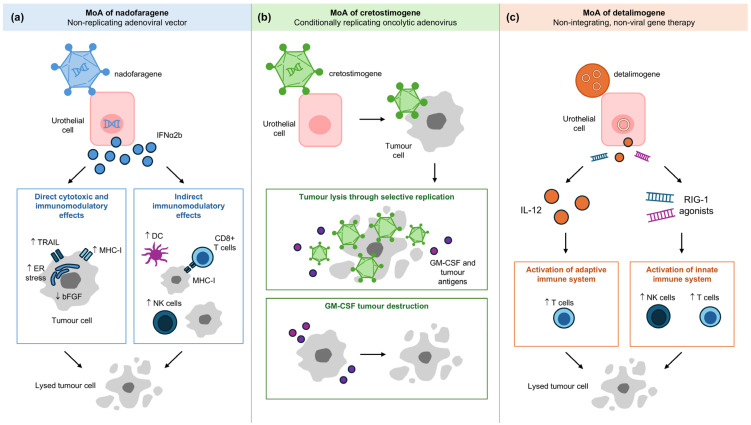
Mechanism of action of (**a**) nadofaragene firadenovec, (**b**) cretostimogene grenadenorepvec and (**c**) detalimogene voraplasmid. DC, dendritic cell; ER, endoplasmic reticulum; GM-CSF, granulocyte-macrophage colony-stimulating factor; IL, interleukin; MHC-I, major histocompatibility complex class I; MoA, mechanism of action; NK, natural killer; RIG-I, retinoic acid-inducible gene I; TRAIL, tumour necrosis factor-related apoptosis-inducing ligand.

**Table 1 cancers-17-03631-t001:** Summary of gene therapies for the treatment of high-risk BCG-unresponsive NMIBC with CIS.

Agent	Delivery Platform	Therapeutic Gene (s) Delivered	Primary Immunologic Goal	Dosing
Nadofaragene firadenovec	Non-replicating adenovirus [11,22]	*IFNα2b*	Enhance innate and adaptive immunity	Intravesical administration Once every 3 months
Cretostimogene grenadenorepvec (CG0070)	Replicating adenovirus [23,24,25]	*GM-CSF*	Induce immunogenic cell death	Intravesical administration 6 weekly doses during induction, followed by 3 weekly maintenance doses at months 3, 6, 9, 12, and 18
Detalimogene voraplasmid (EG-70)	Non-viral plasmid delivery [10,26,27]	*IL-12* *RIG-I*	Prime and expand T-cell and NK-cell response	Intravesical administration Weeks 1, 2, 5, and 6 of a 12-week cycle, for 4 cycles

BCG, *Bacillus Calmette-Guérin*; CIS, carcinoma in situ; GM-CSF, granulocyte-macrophage colony-stimulating factor; IFNα2b, interferon alpha-2b; IL-12, interleukin-12; NK-cell, natural killer cell; NMIBC, non-muscle invasive bladder cancer; RIG-I, retinoic acid-inducible gene I.

**Table 2 cancers-17-03631-t002:** Key efficacy and safety results of gene therapy in high-risk BCG-unresponsive NMIBC with CIS.

Agent	Regulatory Status	Study Phase	Number of Patients	Efficacy			Safety
				Median Follow-Up	CR	Median DoR	
Nadofaragene firadenovec	FDA-approved	Phase III [22,35]	157 ^a^	19.7 mo	3 mo: 53.4% (n = 55/103) 6 mo: 40.8% (n = 42/103) 9 mo: 35.0% (n = 36/103) 12 mo: 24.3% (n = 25/103)	9.69 mo	Grade 3 AEs: 3.8% (n = 6/157) No Grade 4/5 TRAEs Discontinuation due to AEs: 1.9% (n = 3/157)
Cretostimogene grenadenorepvec (CG0070)	FDA Fast Track and Breakthrough Therapy Designation	Phase III [37]	112	22.3 mo	12 mo: 46.4% (n = 51/110) Any time: 75.5% (n = 83/110)	27.9 mo	No Grade ≥ 3 AEs Grade 1/2 TRAEs were transient (median time to resolution of 1 day)
		Phase II (cretostimogene + pembrolizumab) [24]	35	NR	12 mo: 57% (n = 20/35) Any time: 83% (29/35)	Has not been reached (>21 mo)	Safety profile consistent with individual agents No synergistic toxicity observed
Detalimogene voraplasmid (EG-70)	Investigational, not yet approved	Phase I/II (ongoing) [38]	42	NR	Preliminary results Overall: 71% (n = 15/21) 3 mo: 67% (n = 14/21) 6 mo: 47% (n = 8/17)	NR	TRAEs: 47.6% (n = 20/42) All TRAEs of Grade 1/2

^a^ Of the 157 patients enrolled, 107 patients had CIS ± Ta/T1 and 50 patients had high-grade Ta/T1; 6 patients did not meet the definition of BCG-unresponsive NMIBC and therefore were not included in the efficacy analyses. AE, adverse event; BCG, *Bacillus Calmette-Guérin;* CIS, carcinoma in situ; CR, complete response; DoR, duration of response; FDA, Food and Drug Administration; HGRFS, high-grade recurrence-free survival; mo, months; NR, not reported; TRAE, treatment-related adverse event.

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
