# Peer review of "Gene Therapy for BCG-Unresponsive Non-Muscle Invasive Bladder Cancer: Current Evidence and Future Directions"

_cancers, 2025, doi:10.3390/cancers17223631_

Round 1

Reviewer 1 Report

Comments and Suggestions for Authors

Dear Authors,

Firstly, I would like to commend you for addressing such a clinically relevant and timely topic. The management of BCG-unresponsive non-muscle invasive bladder cancer (NMIBC) represents a pressing unmet need in urologic oncology, and your review provides a comprehensive overview of the current landscape of gene therapy. The structure of the manuscript is logical, and the inclusion of both FDA-approved agents and investigational products ensures that readers obtain a balanced update on recent progress in the field.

Secondly, the organization of the review is clear, but there is room to strengthen the critical appraisal of the evidence. Many of the clinical trials you discuss are single-arm and non-comparative, which inevitably limits the generalizability of the findings. While you acknowledge this in part, it would benefit the reader if you explicitly emphasized how these methodological constraints impact clinical decision-making. For instance, highlighting the absence of head-to-head comparisons with radical cystectomy or systemic immunotherapy would provide a more realistic context for interpreting the reported response rates.

Thirdly, the discussion of patient-reported outcomes and quality of life is an important addition, yet it could be expanded with concrete examples. Including which clinical studies have utilized validated instruments such as the EORTC QLQ-NMIBC24 or FACT-Bl-Cys would illustrate how gene therapy trials are (or are not) incorporating patient-centered endpoints. This would make your argument for quality-of-life benefits more persuasive and evidence-driven.

Fourthly, the section on cost and access considerations is highly relevant, but it remains somewhat abstract. Readers would benefit from more concrete details such as preliminary estimates of cost-effectiveness or references to existing health technology assessments. Such data would strengthen the manuscript’s translational value, especially for clinicians and policymakers considering the integration of gene therapy into practice.

Finally, while your figures and tables are useful in summarizing key concepts and outcomes, they could be better tied to practical clinical implications. For example, the mechanisms of action illustrated in Figure 3 could be more explicitly linked to potential predictors of response or resistance. Similarly, Table 2 could be discussed in terms of how efficacy and safety profiles may influence treatment sequencing in real-world settings.

In summary, your review is scientifically sound, up to date, and of considerable educational value. With a stronger emphasis on trial limitations, patient-centered outcomes, and practical implementation issues, it will serve as an excellent resource for both researchers and clinicians working in the field of bladder cancer therapeutics.

Sincerely,

Author Response

Please see the joint document.

Reviewer 2 Report

Comments and Suggestions for Authors

This review highlights current gene therapy approaches for BCG-unresponsive bladder cancer (BCa). The authors describe several promising strategies, including a recombinant adenovirus delivering interferon alpha-2b, an oncolytic adenoviral vector, and a non-viral gene therapy approach delivering interleukin-12 (IL-12) in combination with retinoic acid-inducible gene I (RIG-I). The mechanisms of action for each therapy are discussed, along with key clinical trials supporting their development. Future perspectives in this rapidly evolving field are also presented.

The authors provide basic information on BCa and an overview of non-muscle-invasive BCa treatment algorithms in two figures, which could be very helpful for non-urologist readers of the manuscript.

Overall, this is a comprehensive and up-to-date review that will be of interest to a broad audience, including oncologists, urologists, and basic researchers.

The manuscript includes three figures, two tables, and 56 references—an appropriate number for a review of this scope.

Minor revisions required before acceptance:

  1. The details of detalimogene voraplasmid (EG-70) should be briefly mentioned in the Abstract.
  2. In Figure 1, according to the TNM classification, T4 indicates that the tumor invades other organs, while spread to lymph nodes is classified as N1–N3. This should be corrected. Mentioning of lymph node involvement should also be deleted from the description of T3.
  3. In Figure 3(c), “IL-2” should be corrected to “IL-12.”

Author Response

Please see the joint document.
